# Removal of Linear and Branched Alkylphenols with the Combined Use of Polyphenol Oxidase and Chitosan

**DOI:** 10.3390/polym11060931

**Published:** 2019-05-28

**Authors:** Mitsuki Tsushima, Yuji Kimura, Ayumi Kashiwada, Kazunori Yamada

**Affiliations:** Department of Applied Molecular Chemistry, College of Industrial Technology, Nihon University, 1-2-1 Izumi-cho Narashino, Chiba 275-8575, Japan; tm25128@gmail.com (M.T.); kimura.yuuji@nihon-u.ac.jp (Y.K.); kashiwada.ayumi@nihon-u.ac.jp (A.K.)

**Keywords:** polyphenol oxidase, chitosan, alkylphenols, quinone oxidation, quinone adsorption

## Abstract

Removal of linear and branched alkylphenols with different alkyl chain lengths or different branchings (normal, secondary, and tertiary), some of which are suspected as endocrine disrupting chemicals, from an aqueous medium were investigated through quinone oxidation by polyphenol oxidase (PPO) and subsequent quinone adsorption on chitosan beads or powders at pH 7.0 and 40 °C. PPO-catalyzed quinone oxidation increased with an increase in alkyl chain length of the alkylphenols used. Although a higher PPO dose was required for quinone oxidation of branched alkylphenols, they were completely or mostly removed by quinone adsorption on chitosan beads or powders. The apparent activity of PPO increased by a decrease in quinone concentration. On the other hand, in the homogeneous systems with solutions of chitosan and PPO at pH 6.0, longer reaction times were required to generate insoluble aggregates, and a small amount of quinone derivatives were left in the solution even under optimum conditions. These results support that the two-step reaction, that is, PPO-catalyzed quinone oxidation and subsequent quinone adsorption on chitosan beads or powders, in the heterogeneous system is a good procedure for removing linear and branched alkylphenols from aqueous medium.

## 1. Introduction

Interference exerted by an exogenous substance in the hormonal system is referred to as endocrine disruption, and the substances themselves are known as endocrine-disrupting chemicals (EDCs). EDCs and potential EDCs are found in various materials such as pesticides, additives, or contaminants in food and personal care products. EDCs are also called as xenoestrogens. They mean “foreign” estrogens that have estrogenic effects on a living organism, even though they differ chemically from estrogenic substances internally produced by the endocrine system of an organism. Bisphenol A (BPA) and its derivatives—phthalates, alkylphenols, phytoestrogens, dichloro- diphenyltrichloroethane (DDT), polychlorinated biphenyls (PCBs), and dioxins—are representative xenoestrogens with similar structures to 17β-estradiol in EDCs [1,2].

Alkylphenols have been used as raw materials of alkylphenol polyethoxylate-type nonionic surfactants, antioxidants for plastics, and heat stabilizers for polyvinyl chloride, and they frequently exist in surface waters [3,4]. Alkylphenol ethoxylates undergo degradation by the stepwise loss of ethoxy groups to form shorter ethoxylate homologs, ethoxycarboxylates, and ultimately alkylphenols. Alkylphenols with an alkyl side chain on the para position exhibit weak estrogen-like actions, and para-alkylphenols with longer alkyl chains show higher estrogen receptor binding affinities [5,6]. Studies on the removal or degradation of alkylphenols have been widely performed by chemical procedures, such as electrolysis [7], photolysis [8,9], adsorption [10], nanofiltration [11], ozone treatment [12], and chemical oxidation [13], and by aerobic or anaerobic biological processes [14,15,16]. Some of these procedures often face the following problems that need to be solved: formation and/or residue of by-products, high costs, low efficiency, and limited specificity for targeted compounds.

Alternatively, so far, we have reported the removal of phenol compounds with oxidoreductases [17,18,19,20,21,22,23]. Peroxidases catalyze the conversion of phenol compounds into the corresponding phenoxy radicals in the presence of hydrogen peroxide (H_2_O_2_). Subsequently, water-insoluble oligomers are generated by alternately repeating the nonenzymatic coupling reaction and enzymatic radical formation [17,18,19]. Alternatively, tyrosinases catalyze quinone oxidation of phenol compounds [24]. Many phenol compounds are removed through the reaction of enzymatically generated quinone derivatives with amino group-containing polymers, such as chitosan and polyethyleneimine [25,26]. In these articles, we reported some findings, such as the decrease in enzyme dose, the shortage in removal time, and the enhancement of removal efficiency with chitosan beads, in the heterogeneous system [20,21,22,23]. Mushroom tyrosinase oxidizes many linear and branched *p*-alkylphenols into the corresponding quinone derivatives. However, 4-*tert*-butylphenol (4TBP) and 4-*tert*-pentyl phenol (4TPenP) undergo no quinone oxidation in the absence of H_2_O_2_ by mushroom tyrosinase. García-Carmona et al. reported that mushroom tyrosinase catalyzed quinone oxidation of 4TBP in the presence of H_2_O_2_ [27,28]. In addition, there was another problem to be solved. The tyrosinase dose increased for quinone oxidation of alkylphenols with longer alkyl chain lengths [20,23]. An alternative oxidoreductase, polyphenol oxidase (PPO), has been reported to catalyze quinone oxidation of phenol compounds [29,30,31,32]. They were removed with the combined use of PPO and chitosan [33,34].

In this study, a reaction system constructed with the combined use of PPO-catalyzed quinone oxidation and the heterogeneous reaction with chitosan is applied for the removal of linear and branched alkylphenols with different alkyl chain lengths and branchings. First, enzymatic quinone oxidation of alkylphenols is investigated in close relation with alkyl chain length and branching (normal, secondary, or tertiary) of the target alkylphenols. In addition, quinone adsorption is followed up in the heterogeneous system on chitosan beads or powders and in the homogeneous system with solutions of PPO.

## 2. Experimental

### 2.1. Materials and Chemicals

Linear and branched alkylphenols with different alkyl chain lengths and branchings that were used are listed in Table 1. 4TPenP was purchased from Wako Pure Chemical (Osaka, Japan), and the other alkylphenols were purchased from Tokyo Chemical Industry (Tokyo, Japan). These were used as received. PPO (EC 1.14.18.1) was purchased from Worthington Biochemical (Lakewood, NJ, USA). The nominal specific activity of PPO used was 925 U/mg. A crosslinked chitosan bead and chitosan powders of three different sizes were used as an adsorbent for quinone. The chitosan beads, Chitopearl AL-01 (particle size: 70–210 μm, specific surface area: 70–100 m^2^/g, and amino group content: 0.25 mmol/cm^3^) were obtained from Fuji Spinning (Tokyo, Japan). Water content of the chitosan beads used was determined to be 92.5 wt% in water. The chitosan powders were kindly provided from Dainichiseika Color & Chemicals (Tokyo, Japan), and they had diameters of 10–42, 42–74, and 74–100 μm (degree of deacetylation = 85%). Viscosity (nominal value) of the used chitosan was <500 mPa·s at 20 °C (concentration = 0.5 wt%, and solvent = 0.5 wt% acetic acid).

### 2.2. Removal of Alkylphenols in the Heterogeneous Reaction

Stock solutions of alkylphenols and PPO were prepared in a pH 7.0 phosphate buffer (ionic strength = 0.01 M) at prescribed concentrations [33,34]. Chitosan beads were washed with buffer several times and then preserved in buffer before adsorption experiments. In a 50 cm^3^ Erlenmeyer flask, a PPO solution was added to each alkylphenol solution of 30 cm^3^, and then the solutions were diluted with buffer to a final volume of 40 cm^3^ to achieve the initial alkylphenol concentrations shown in Table 1. The initial concentration was adjusted to 0.30 mM for alkylphenols with an alkyl chain length of 1 to 6 [17,20,21,23]. The initial concentration of 4-*n*-hexylphenol (4NHexP) was adjusted to 0.20 mM because the solubility of 4NHexP in a pH 7.0 buffer was low. For 4-*n*-heptylphenol (4NHepP), 4-*n*-octylphenol (4-NOP), 4-*tert*-octylphenol (4TOP), and 4-*n*-nonylphenol (4NNP), the initial concentration was adjusted to 0.10 mM by using a pH 7.0 buffer containing DMSO. DMSO concentration in these reaction solutions is shown in Table 1.

When the UV-visible spectra of the reaction solutions were measured over the reaction time, the peak appeared at 400 nm. Therefore, absorbance at 400 nm was used as a measure of quinone generation [17,20,21,23]. For removal experiments, 4.0 cm^3^ of chitosan beads were added to each alkylphenol solution with a bullet, and then the enzymatic reaction was initiated by adding PPO to the solutions [20,21,22,23]. Three kinds of chitosan powders were used in place of chitosan beads for quinone adsorption experiments. Since chitosan beads of 4.0 cm^3^ in the swollen state consisted of 0.30 g chitosan and 3.70 g water, according to the water content value, 0.30 g of chitosan powders was dispersed in alkylphenol solutions containing PPO.

### 2.3. Quantitative Assay of Alkylphenols

The concentration of alkylphenols remaining in the solutions was determined by Hitachi L-7000 high-performance liquid chromatography (HPLC) combined with a UV-visible spectrophotometer and an integrator (Hitachi L-7420) [17,20,21,23]. A reverse phase column, Inertsil ODS-2 (particle size: 5 μm, 4.6 mm i.d. × 15 cm), was used, and the composition of the aqueous acetonitrile solution as a mobile phase was adjusted so as to set the retention time of each alkylphenol between 7 and 9 min. A solution of precisely 20 mm^3^ was injected into the HPLC system with a microsyringe, and the mobile phase ran at 1.0 cm^3^/min. The conversion percent value was calculated from the peak areas before the enzymatic reaction and at time *t* according to Equation (1).
(1)$$\mathrm{Conversion}\;\%=\;\frac{{\mathrm{Area}}_{0}-{\mathrm{Area}}_{t}}{{\mathrm{Area}}_{0}}\times 100$$

### 2.4. Removal of Alkylphenols in the Homogeneous System

Homogeneous removal experiments were performed as a control. Chitosan solutions of 1.0–1.5 *w*/*v*% were prepared by dispersing chitosan powders in distilled water and intermittently adding 2 M HCl solution dropwise with constant stirring for at least 24 h to maintain a pH value of 3–4. Then, insoluble parts were removed by vacuum filtration with a G3 glass filter. The obtained chitosan solutions were diluted with distilled water to an amino group concentration of 0.15–30 mM based on the weight concentration of the chitosan solution and the degree of deacetylation of the used chitosan powder [22]. The stock solutions of alkylphenols and PPO were prepared in a pH 6.0 phosphate buffer (ionic strength = 0.01 M) at prescribed concentrations.

To the solutions of butylphenols and pentylphenols, the solutions of PPO and chitosan were added so as to reach an alkylphenol concentration of 0.30 mM. The molar ratio of chitosan’s amino group to alkylphenol ranged from 0.05 to 10.0, which corresponded to an amino group concentration of 0.015–3.0 mM. The solutions were stirred at 40 °C, and the absorbance of the solutions was measured at predetermined time intervals. In the case where insoluble aggregates were generated in the solution, the absorbance of the filtrates was measured after removal with a 5C filter paper.

## 3. Results and Discussion

### 3.1. Polyphenol Oxidase (PPO) Concentration Dependence of Quinone Oxidation

Effects of enzyme concentration on PPO-catalyzed quinone oxidation of 14 kinds of linear and branched alkylphenols were investigated at pH 7.0 and 40 °C (Table 1). These values were determined as the optimum conditions of PPO used in this study for BPA and its derivatives in our previous articles [33,34]. Therefore, these optimum conditions were used for PPO-catalyzed quinone oxidation in this study. Figure 1 shows changes in the conversion percent value at a reaction time of 60 min as well as the initial velocity of conversion into the corresponding catechols with the PPO concentration for PPO-catalyzed quinone oxidation of linear alkylphenols with alkyl chain lengths of 1 to 6. Quinone oxidation of these linear alkylphenols increased with an increase in PPO concentration irrespective of the alkyl chain length. The initial velocity of conversion into the corresponding catechols, or the cresolase activity, was proportional to the PPO concentration. The decrease in alkylphenol concentration meant that each alkylphenol was converted into the corresponding catechols by PPO in the presence of molecular oxygen [27,28,35]. In addition, the initial velocity linearly increased against the PPO concentration.

The appearance of the absorbance at 400 nm showed that the alkylphenols used underwent PPO-catalyzed quinone oxidation in the same manner as the case when mushroom and *Aspergillus* tyrosinases were used [20,21,22,23,33,34]. This meant that the catechol intermediates were converted into the corresponding quinone derivatives by the catecholase activity of PPO. In addition, the absorbance more slowly increased against the reaction time than the conversion percent value did against the reaction time. Quinone oxidation of alkylphenols by PPO occurred by a two-step enzymatic reaction: cresolase activity to oxidize a phenol compound into the corresponding catechol compound and catecholase activity to oxidize the catechol compound into the corresponding quinone derivative [36,37]. Since catechols are colorless, a decrease in alkylphenol concentration against the reaction time is referred to as the velocity of conversion of alkylphenols into the corresponding catechols. Thus, it was found from the above results that the velocity of conversion into catechols by cresolase activity was higher than that of conversion into the corresponding quinone derivatives by catecholase activity.

The use of enzymes with higher reaction specificities is desirable for the treatment of phenol compounds by an oxidoreductase. We reported that an increase in alkyl chain length of alkylphenols, or between two phenol groups of BPA derivatives, resulted in an increase in enzyme dose for the treatment of alkylphenols [17,20,21,23] or BPA and its derivatives [18,19,22] by mushroom tyrosinase [20,22], *Aspergillus oryzae* tyrosinase [23], horseradish peroxidase [17,18], and soybean peroxidase [19]. The specific initial velocity of PPO used in this study increased from 1.14 × 10^−3^ μmol/min·g for *p*-cresol to 4.29 × 10^−3^ μmol/min·g for 4NHexP. Here, it should be noted that *n*-alkyl phenols with alkyl chain lengths of 1 to 6 underwent quinone oxidation at lower concentrations than mushroom tyrosinase and *Aspergillus oryzae* tyrosinase [17,20,23].

Next, quinone oxidation of branched alkylphenols was investigated at pH 7.0 and 40 °C. Figure 2 shows changes in the conversion percent value at 60 min and the initial velocity with PPO concentrations for 4-*iso*-propylphenol (4IProP), 4-*sec*-buthylphenol (4SBP), 4TBP, and 4TPenP. Quinone oxidation of these branched alkylphenols also increased with PPO concentration in the same manner as the corresponding linear alkylphenols. In addition, comparison of Figure 1 and Figure 2 showed that a higher dose of PPO was required for quinone oxidation of branched alkylphenols than for that of corresponding linear alkylphenols. The initial velocities for linear alkylphenols were lower than that for branched alkylphenols, for example, 4-*n*-propylphenol (4NProP) > 4IProP and 4-*n*-pentylphenol (4NPenP) > 4TPenP. For butylphenols, the initial velocity for 4-*n*-buthylphenol (4NBP) had the highest value and was followed by 4SBP and 4TBP (4NBP > 4SBP > 4TBP). These results revealed that PPO catalyzed quinone oxidations of these branched alkylphenols at lower concentrations than mushroom tyrosinase [20,23].

In addition, H_2_O_2_ is required for quinone oxidation of 4TBP by mushroom tyrosinase [27], whereas PPO-catalyzed quinone oxidation occurs without H_2_O_2_. Figure 3 shows changes in the conversion percent value at 60 min and the initial velocity with PPO concentrations for 4NHepP, 4NOP, 4TOP, and 4NNP at pH 7.0 and 40 °C. These alkylphenols also underwent quinone oxidation by PPO without H_2_O_2_. In particular, it was found that quinone oxidation of 4TOP was much slower than that of 4NOP.

### 3.2. PPO-Catalyzed Quinone Oxidation and Subsequent Quinone Adsorption on Chitosan Beads

Since it was found in the above section that the linear and branched alkylphenols used were successfully quinone-oxidized by PPO without H_2_O_2_, quinone oxidation by PPO was performed in the presence of chitosan beads for their adsorptive removal. Figure 1, Figure 2 and Figure 3 also showed changes in the conversion percent value at 60 min and the initial velocity with the PPO concentration in the presence of chitosan beads at 0.10 cm^3^/cm^3^ (shaded plots). Solutions of these alkylphenols turned brown by enzymatic quinone generation in the absence of chitosan beads. On the other hand, in the presence of chitosan beads, an increase in the absorbance was highly suppressed by adsorption of enzymatically generated quinone derivatives on the chitosan beads [20,22,23]. This indicated that these alkylphenols were removed by quinone adsorption on chitosan beads. Quinone adsorption occurs through a chemical reaction of enzymatically generated quinone derivatives with chitosan’s amino groups. This mechanism was explained in detail in ours and Payne’s articles [20,22,23,26]. Results of removing the 14 kinds of alkylphenols used are summarized in Table 2. Quinone oxidation in the presence of chitosan beads was faster than in the absence of chitosan beads, and the alkylphenol concentration decreased at shorter reaction times. Also, these alkylphenols were completely or mostly quinone-oxidized in the presence of chitosan beads. Absorbance slightly increased in the beginning of the reaction and was followed by a decrease to zero or nearly zero by adsorption on the chitosan beads. As a result, the alkylphenols used were highly removed at a reaction time of 3 to 5 h.

Many articles described that when quinone derivatives enzymatically generated from various phenol compounds reacted with amino groups of chitosan, a peak appeared at the wavelength of about 420 nm. Chitosan films immersed in the solutions containing a phenol compound and PPO turn dark-brownish. On the other hand, a cellulose film did not stain at all [20,23,26,27,33,34]. This indicated that the enzymatically generated quinone derivatives successfully reacted with chitosan’s amino group but did not react with the hydroxyl group. Removal of various alkylphenols performed in this study was also based on the same reaction mechanism.

In addition, it can be seen from Figure 1, Figure 2 and Figure 3 that higher conversion percent values were obtained in the presence of chitosan beads than in the absence of chitosan beads. The ratio of the conversion percent value in the presence of chitosan beads to that in the absence of chitosan beads was calculated and defined as the initial velocity ratio. Figure 4 shows changes in the initial velocity ratio with the PPO concentration for the linear alkylphenols used. The initial velocity ratio decreased with an increase in PPO concentration. Inhibition of enzymatic activity was found to be suppressed through a decrease in quinone concentration in the reaction solutions by quinone adsorption. This behavior was quite evident at lower PPO concentrations.

Next, Figure 5 shows changes in the initial velocity ratio with the PPO concentration for the branched alkylphenols used. The initial velocity ratio for branched alkylphenols gradually decreased with an increase in enzyme concentration. As shown in Figure 4, the initial velocity ratio for the linear alkylphenols ranged from 1.1 to 2.6, whereas those for the branched alkylphenols ranged from 2.4 to 4.4. This indicated that the enzymatic activity of PPO apparently increased by dispersing chitosan beads in the alkylphenol solutions, and in particular, this behavior was more prominently observed for the branched alkylphenols. Inhibition of enzyme activity was found to be more highly suppressed by quinone adsorption on the chitosan beads for the branched alkylphenols.

Chitosan beads were dispersed in alkylphenol solutions without PPO at pH 7.0 and 40 °C, and then the solutions were constantly stirred. Figure 6 shows the amount of alkylphenols physically or electrostatically adsorbed on the chitosan beads after stirring for 6 and 24 h. The adsorbed amount increased with an increase in alkyl chain length of the alkylphenols used and/or the stirring time. However, a considerable amount of alkylphenols were left in the solutions. This result emphasized that although PPO dose was required to be adjusted, depending on the length and branching of alkyl chains, the above-mentioned two-step procedure was quite effective in removing linear and branched alkylphenols from the aqueous medium.

### 3.3. PPO-Catalyzed Quinone Oxidation and Subsequent Quinone Adsorption on Chitosan Powder

Since the alkylphenols used were removed through quinone adsorption on chitosan beads, the removal of 4NBP, 4SBP, 4TBP, 4NPenP, and 4TPenP by the above-mentioned two-step procedure was investigated with three kinds of chitosan powders in place of chitosan beads. Figure 7 shows the time course in the absorbance and conversion percent value for 4NPenP solutions containing PPO, in which the three kinds of chitosan powders or chitosan beads were dispersed. 4TPenP was also successfully removed through quinone adsorption on chitosan powders. As the diameter of the chitosan powders used was lower, the conversion percent value increased at shorter reaction times. Such behavior was also observed for 4NBP, 4SBP, 4TBP, and 4TPenP (not shown). The minimum absorbances and conversion percent values were summarized in Table 3 in the case where PPO was added to these five kinds of alkylphenol solutions in the presence of chitosan powders or chitosan beads at pH 7.0 and 40 °C. On the supposition that the chitosan powders used were perfect spheres with an average diameter of 26, 58, and 87 μm, and chitosan had a density of 1.0 g/cm^3^ in the dry state, the specific surface areas of the chitosan powders with the diameters of 10–42, 42–74, and 74–100 μm were calculated to be 0.231, 0.103, and 0.069 m^2^/g, respectively [22]. These values were much smaller than the nominal specific surface area value (70–100 m^2^/g) of the chitosan beads used. These five kinds of alkylphenols were highly removed by quinone adsorption on chitosan powders with diameters of 10–42 and 42–74 μm. It was found from these results that if the chitosan powders were used in the heterogeneous system at pH 7.0, these alkylphenols were more effectively removed than the chitosan beads. In other words, it was especially important that enzymatically generated quinone derivatives reacted with chitosan in the heterogeneous system. However, it was quite difficult to determine the amount of amino groups present on the surfaces of the chitosan powders, because chitosan was insoluble at pH 7.0.

### 3.4. Removal with PPO and Chitosan in the Homogeneous System

PPO was added to the solutions of 4NBP, 4SBP, 4TBP, 4NPenP, and 4TPenP containing chitosan at different amino group concentrations in a pH 6.0 buffer at 40 °C. The reason for the use of a pH 6.0 buffer was simply because chitosan was insoluble at pH values higher than 6.0, and the degree of protonation was minimized. Figure 8 shows changes in the absorbance with the amino group concentration or the molar ratio of chitosan amino groups to alkylphenols. When enzymatically generated quinone derivatives reacted with amino groups of chitosan [20,22,25,26], water-insoluble dark-brownish aggregates were generated. Absorbance decreased with an increase in amino group concentration to reach the minimum values. However, when the amino group concentration was further increased, the absorbance sharply increased for 4NBP, 4SBP, and 4NPenP and gradually increased for 4TBP and 4TPenP.

At low amino group concentrations, free quinone derivatives were left in the solution, and aggregate scarcely formed because chitosan was in short supply. On the other hand, since many free amino groups were left on a chitosan chain, and some chitosan chains remained soluble in the solution at higher amino group concentrations, a high absorbance was empirically observed for 4NBP, 4SBP, and 4NPenP. The amino group concentrations at which minimum absorbance was observed were determined as the optimum value, and at these amino group concentrations, the time course of absorbance was measured. Figure 9 shows the time course of absorbance at amino group concentrations for 4NBP, 4SBP, 4TBP, 4NPenP, and 4TPenP. The used amino group concentration was 0.225 mM for 4NBP, 4SBP, and 4NPenP and 0.45 mM for 4TBP and 4TPenP. Absorbance increased by the generation of quinone derivatives at a short reaction time depending on the length and branching of the alkyl chain. After minimum absorbance, the absorbance sharply decreased for 4NBP and 4NPenP and gradually decreased for 4SBP, 4TBP, and 4TPenP against the reaction time by the generation of aggregates. The sharp decrease in the absorbance suggested that the quinone derivatives enzymatically generated from 4NBP and 4NPenP successfully reacted with chitosan’s amino groups. On the other hand, the reactions of quinone derivatives generated from 4SBP, 4TBP, and 4TPenP with chitosan’s amino groups were considered to be considerably slow. In addition, another reason was that quinone derivatives generated from 4TBP and 4TPenP were less stable in the aqueous medium [27]. As shown in Table 3, these five kinds of alkylphenols were almost completely removed for 3 to 5 h by quinone oxidation and subsequent quinone adsorption on chitosan beads, whereas in the homogeneous reaction, formation of aggregates was not observed at theses reaction times (Figure 9). Consequently, this comparison emphasizes that alkylphenols can be removed in much shorter reaction times for the heterogeneous two-step reaction with PPO and chitosan beads than for the homogeneous reaction.

## 4. Conclusions

In this study, removal of linear and branched alkylphenols was investigated with the combined use of PPO-catalyzed quinone oxidation and subsequent reaction of enzymatically generated quinone derivatives with chitosan in different forms of porous beads, powders, and solutions. At pH 7.0 and 40 °C, the alkylphenols used were successfully converted into the corresponding quinone derivatives by PPO. Quinone oxidation occurred at lower PPO concentrations for alkylphenols with shorter alkyl chains.

When PPO was added to alkylphenol solutions in the presence of chitosan beads, the alkylphenols were removed by chemical adsorption, or chemisorption, of enzymatically generated quinone derivatives on chitosan beads, and the removal time was shortened by increasing PPO concentration. Here, the activity of PPO apparently increased as a result of the decrease in quinone concentration by adsorption on chitosan beads. This behavior is quite favorable for this removal system. PPO catalyzed quinone oxidation of alkylphenols without H_2_O_2_ at lower concentrations compared to mushroom tyrosinase, although a higher PPO dose was required for quinone oxidation of branched alkylphenols than for that of the corresponding linear ones. Consequently, it should be noted that 4SBP, 4TBP, 4TPenP, and 4TOP, suspected as endocrine-disrupting chemicals, were also successfully removed for 1.5–4 h. Even if chitosan powders were dispersed in place of chitosan beads in the alkylphenol solution containing PPO at pH 7.0, alkylphenols were removed by quinone adsorption, and removal time was shortened as the diameter of the chitosan powder decreased.

On the other hand, in the homogeneous system with solutions of PPO and chitosan at pH 6.0, the alkylphenols used were removed through the formation of aggregates by the chemical reaction of enzymatically generated quinone oxidation with chitosan’s amino groups. However, much longer removal times were required compared to heterogeneous systems with chitosan beads or powders. These results emphasize the fact that quinone adsorption in the heterogeneous system is quite effective in removing alkylphenols from the aqueous medium. In addition, we can safely say that an alternative use of chitosan, which is produced in large amounts by deacetylation of chitin present in the shells of crustaceans such as crabs and prawns, can be found out in this study. From now on, we will perform immobilization of PPO on an insoluble solid carrier to investigate removal of alkylphenols by immobilized PPO and repetitive use of immobilized PPO.

## Figures and Tables

**Figure 1 polymers-11-00931-f001:**
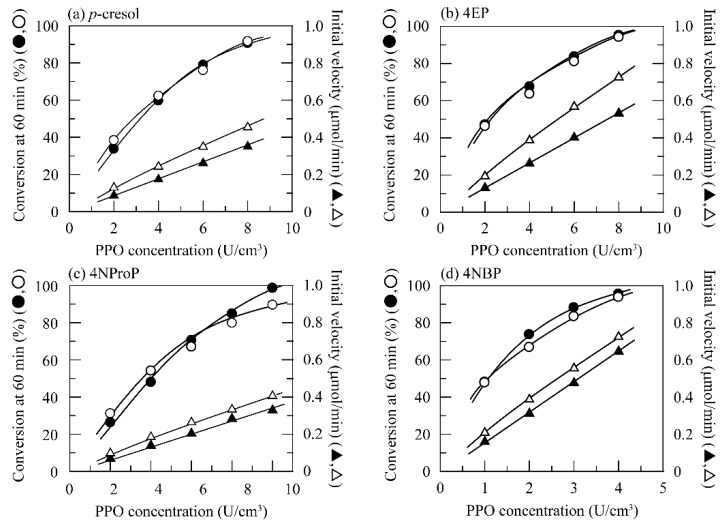
Changes in the conversion percent value at 60 min (circle) and the initial velocity (triangle) with the polyphenol oxidase (PPO) concentration for PPO-catalyzed quinone oxidation of (**a**) *p*-cresol, (**b**) 4EP, (**c**) 4NProP, (**d**) 4NBP, (**e**) 4NPenP, and (**f**) 4NHexP at pH 7.0 and 40°C in the presence (open) or absence (shaded) of chitosan beads at 0.10 cm^3^/cm^3^.

**Figure 2 polymers-11-00931-f002:**
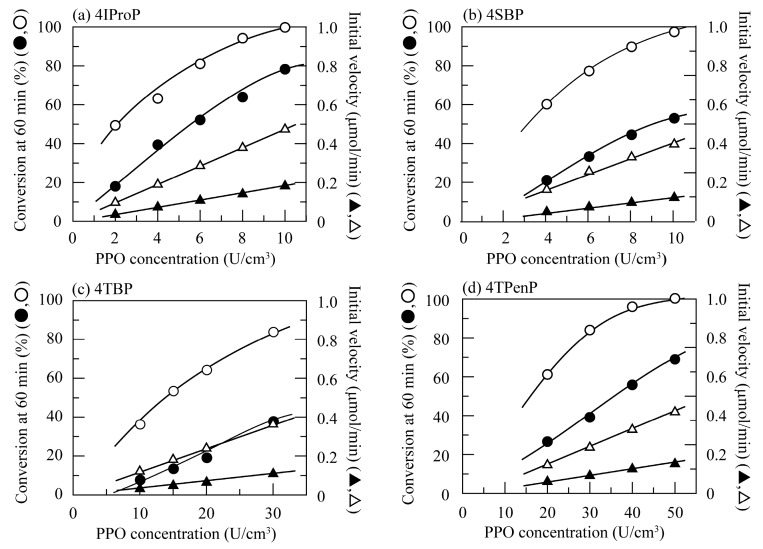
Changes in the conversion percent value at 60 min (circle) and the initial velocity (triangle) with the PPO concentration for PPO-catalyzed quinone oxidation of (**a**) 4IProP, (**b**) 4SBP, (**c**) 4TBP, and (**d**) 4TPenP at pH 7.0 and 40 °C in the presence (open) or absence (shaded) of chitosan beads at 0.10 cm^3^/cm^3^.

**Figure 3 polymers-11-00931-f003:**
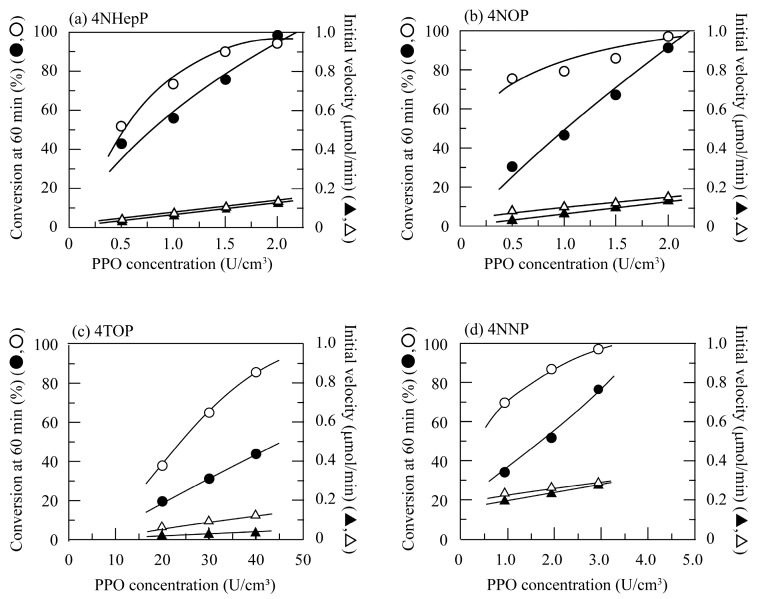
Changes in the conversion percent value at 60 min (circle) and the initial velocity (triangle) with the PPO concentration for PPO-catalyzed quinone oxidation of (**a**) 4NHepP, (**b**) 4NOP, (**c**) 4TOP, and (**d**) 4NNP at pH 7.0 and 40 °C in the presence (open) or absence (shaded) of chitosan beads at 0.10 cm^3^/cm^3^.

**Figure 4 polymers-11-00931-f004:**
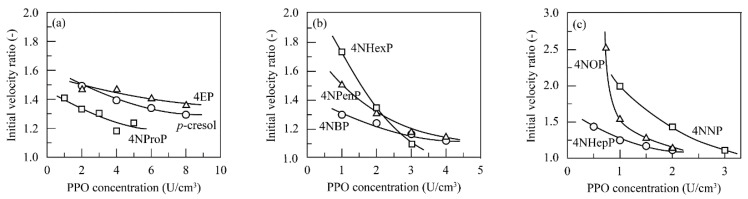
Changes in the initial velocity ratio with the PPO concentration for (**a**) *p*-cresol (○), 4EP (△), and 4NProP (□); (**b**) 4NBP (○), 4NPenP (△), and 4NHexP (□); and (**c**) 4NHepP (○), 4NOP (△), and 4NNP (□) at pH 7.0 and 40 °C.

**Figure 5 polymers-11-00931-f005:**
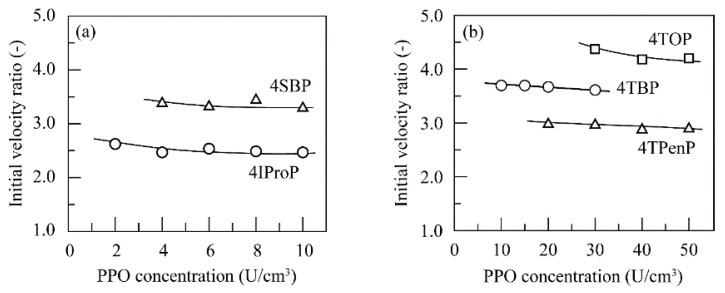
Changes in the initial velocity ratio with the PPO concentration for (**a**) 4IProP (○) and 4SBP (△); and (**b**) 4TBP (○), 4TPenP (△), and 4TOP (□) at pH 7.0 and 40 °C.

**Figure 6 polymers-11-00931-f006:**
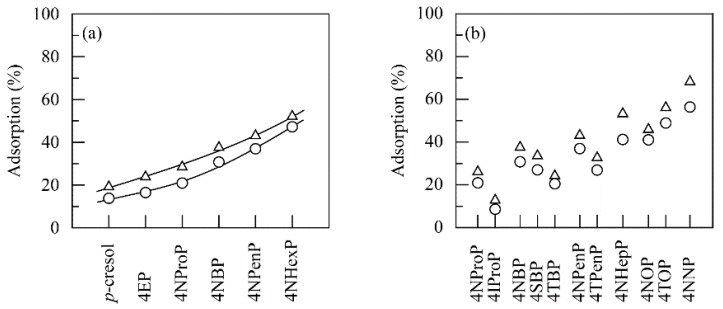
Removal of linear (**a**,**b**) and branched (**b**) alkylphenols on chitosan beads in the absence of PPO for stirring times of 6 (○) and 24 (△) h at pH 7.0 and 40 °C.

**Figure 7 polymers-11-00931-f007:**
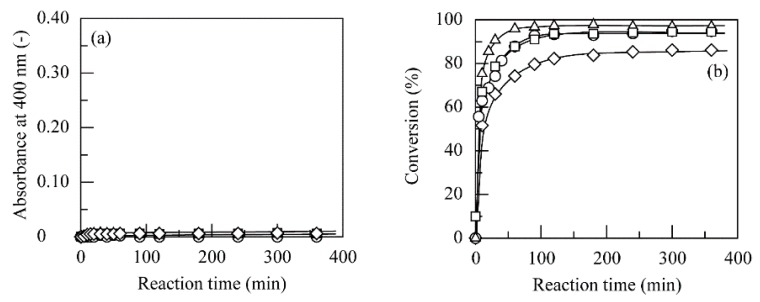
Time course of the (**a**) absorbance at 400 nm and (**b**) conversion percent value for removal of 4NPenP through PPO-catalyzed quinone oxidation and subsequent quinone adsorption on chitosan beads (○) and chitosan powders with diameters 10–42 (△), 42–74 (□), and 74–100 (◇) μm at pH 7.0 and 40 °C.

**Figure 8 polymers-11-00931-f008:**
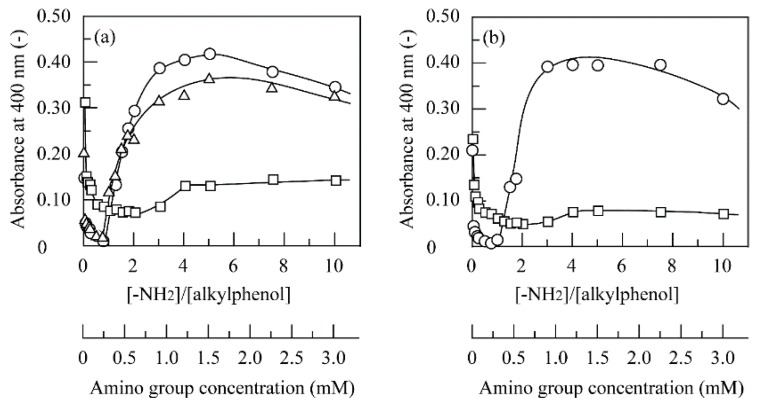
Changes in the absorbance at 400 nm with the amino group concentration for removal of (**a**) 4NBP (○), 4SBP (△), and 4TBP (□); and (**b**) 4NPenP (○) and 4TPenP (□) through PPO-catalyzed quinone oxidation and subsequent homogeneous quinone reaction with chitosan in the presence of PPO at 3.0 U/cm^3^ for 4NBP, 8.0 U/cm^3^ for 4SBP, 20 U/cm^3^ for 4TBP, 2.0 U/cm^3^ for 4NPenP, and 30 U/cm^3^ for 4TPenP at pH 6.0 and 40 °C.

**Figure 9 polymers-11-00931-f009:**
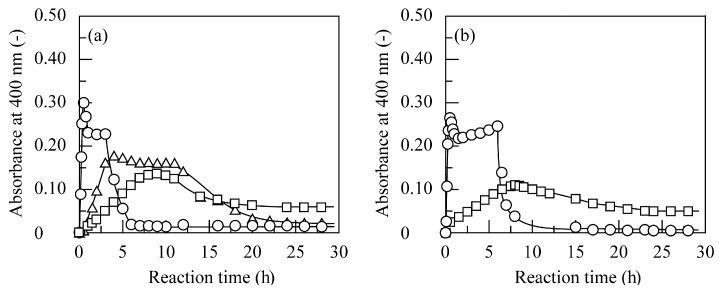
Time course of the absorbance at 400 nm for removal of (**a**) 4NBP (○), 4SBP (△), and 4TBP (□); and (**b**) 4NPenP (○) and 4TPenP (□) through PPO-catalyzed quinone oxidation and subsequent generation of insoluble aggregates in the presence of PPO at 3.0 U/cm^3^ for 4NBP, 8.0 U/cm^3^ for 4SBP, 20 U/cm^3^ for 4TBP, 2.0 U/cm^3^ for 4NPenP, and 30 U/cm^3^ for 4TPenP and chitosan at amino group concentrations of 0.225 mM for 4NBP, 4SBP, and 4NPenP and 0.45 mM for 4TBP and 4TPenP at pH 6.0 and 40 °C.

**Table 1 polymers-11-00931-t001:** Preparation of the solutions of linear and branched alkylphenols used in this study.

Alkyl Chain, R	Abbreviation	Initial Concentration (mM)	Solvent Composition ^(1)^
–CH_3_	*p*-cresol	0.30	buffer
–CH_2_CH_3_	4EP	0.30	buffer
–(CH_2_)_2_CH_3_	4NProP	0.30	buffer
–CH(CH_3_)CH_3_	4IProP	0.30	buffer
–(CH_2_)_3_CH_3_	4NBP	0.30	buffer
–CH(CH_3_)CH_2_CH_3_	4SBP	0.30	buffer
–C(CH_3_)_3_	4TBP	0.30	buffer
-(CH_2_)_4_CH_3_	4NPenP	0.30	buffer
–C(CH_2_CH_3_)(CH_3_)_2_	4TPenP	0.30	buffer
–(CH_2_)_5_CH_3_	4NHexP	0.20	buffer
–(CH_2_)_6_CH_3_	4NHepP	0.10	11.25 vol%
–(CH_2_)_7_CH_3_	4NOP	0.10	26.66 vol%
–C(CH_3_)_2_CH_2_C(CH_3_)_3_	4TOP	0.10	7.5 vol%
–(CH_2_)_8_CH_3_	4NNP	0.10	31.5 vol%

^(1)^ The DMSO concentration in the reaction solution.

**Table 2 polymers-11-00931-t002:** Removal of linear and branched alkylphenols through the PPO-catalyzed quinone oxidation and subsequent quinone adsorption on chitosan beads (0.10 cm^3^/cm^3^) at pH 7.0 and 40 °C.

Alkylphenol	[PPO] (U/cm^3^)	Absence of Chitosan Beads				Presence of Chitosan Beads			
		Max Abs.	Time	Conversion	Time	Min Abs.	Time	Conversion	Time
			(min)	(%)	(min)		(min)	(%)	(min)
*p*-cresol	8	0.278	50	96.5	360	0.02	120	100	300
4EP	8	0.289	30	100	180	0.03	120	99.3	180
4NProP	5	0.254	60	100	180	0.03	180	100	300
4IProP	10	0.339	90	100	180	0.04	180	100	180
4NBP	4	0.285	40	100	180	0.02	180	100	180
4SBP	10	0.315	120	99.3	240	0.02	180	99.3	240
4TBP	30	0.327	240	99.7	300	0.03	240	100	180
4NPenP	4	0.309	30	100	180	0	120	99.3	180
4TPenP	50	0.335	180	100	240	0	120	100	90
4NHexP	3	0.183	30	99.1	180	0	60	100	180
4NHepP	2	0.076	60	97.5	240	0	120	98.4	180
4NOP	2	0.112	180	97.3	180	0	-	97.0	180
4TOP	40	0.10	240	100	360	0	120	100	240
4NNP	2	0.08	240	93.3	240	0	-	97.3	180

**Table 3 polymers-11-00931-t003:** Removal of 4NBP, 4SBP, 4TBP, 4NPenP, and 4TPenP through PPO-catalyzed quinone oxidation and subsequent quinone adsorption on chitosan beads and chitosan powders with diameters 10–42, 42–74, and 74–100 μm at pH 7.0 and 40 °C.

Alkylphenol	[PPO] (U/cm^3^)	Type of Chitosan	Size (μm)	Reaction Time (min)	Min Abs.	Reaction Time (min)	Maximum Conversion
4NBP	3.0	chitosan beads		50	<0.01	240	94.0
			10–42	20	<0.01	240	92.7
		chitosan powder	42–74	30	0.01	240	91.7
			74–100	60	0.01	360	92.3
4SBP	8.0	chitosan beads		60	0.02	300	97.8
			10–42	50	0.01	120	97.8
		chitosan powder	42–74	120	0.03	180	93.0
			74–100	120	0.03	300	88.0
4TBP	20.0	chitosan beads		180	0.03	300	94.0
			10–42	180	0.06	180	100.0
		chitosan powder	42–74	360	0.05	240	98.2
			74–100	360	0.03	240	95.2
4NPenP	20.0	chitosan beads		20	0	300	94.8
			10–42	20	<0.01	90	97.6
		chitosan powder	42–74	40	<0.01	120	93.4
			74-100	60	<0.01	300	85.0
4TPenP	20.0	chitosan beads		240	0	240	100
			10–42	120	0	180	100
		chitosan powder	42–74	180	0	180	100
			74–100	180	0	90	100

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
