# Peer review of "Removal of Linear and Branched Alkylphenols with the Combined Use of Polyphenol Oxidase and Chitosan"

_polymers, 2019, doi:10.3390/polym11060931_

Reviewer 1 Report

The authors describe an approach to remove alkylphenols – known polluting agents – from water. Removal is achieved by first oxidizing the phenols with an enzyme (polyphenol oxidase) to the corresponding quinones and then reacting them with deprotonated chitosan.

The work is interesting and well conducted. However, there are some points that should be refined before publication.

In general, both language and style would benefit from proper English editing.

The manuscript is quite wordy and full of information, often impairing its readability. The authours are encouraged to go directly to the core concepts. But I much appreciated the amount of experimental details given, which is very important for reproducibility.

There are some mistakes and misprints throughout the manuscript. Some examples: pag.1 , line 13 “enhanced with an increase” and line 15 “completely or highly removed”: rephrase; pag. 1, lines 36-38: better to shift the paragraph after the discussion of the uses of alkylphenols; pag. 2, line 72: “was used to emphasize the availability of this procedure”: rephrase; pag. 3, line 102: what does the expression “with a bullet for titration” means? Pag. 5, line 104: the authors mention a “chitosan film”, the preparation of which is not described in the Experimental section: is this a literature reference or an actual experiment? If so, the authors should provide more details.

TITLE: ”adsorptive” is too generic (see the following discussion) and  “oxidoreductase” is redundant. I suggest rephrasing the title as: “Removal of linear and branched alkylphenols with the combined use of polyphenol oxidase and chitosan”.

RESULTS: I am not completely convinced about the “adsorption” of quinones on chitosan. Adsorption suggests a physical mechanism, while a chemical reaction (thus chemisorption) between quinones and the primary amine groups of chitosan seems here the most plausible mechanism at play. Especially considering that the efficiency of removal is reduced when the amine groups of chitosan are protonated i.e. are less reactive. The authors should clarify this point, taking into account that already in the Introduction (lines 53-55) they stated that “Many phenol compounds are removed through the reaction of  enzymatically generated quinone derivatives with amino group-containing polymers, such as chitosan and polyethyleneimine” and also lines 206-207: “This indicates that the enzymatically generated quinone derivatives successfully react with chitosan’s amino group but do not react with the hydroxyl group. The removal of various alkylphenols performed in this study is also based on this reaction mechanism”.

Pag. 6, lines 245-250: while I appreciate the surface area calculations performed, it would be much better if the authors could provide BET results for the chitosan powders and beads. Lines 253-255: Since chitosan is insoluble at pH 7.0, it is especially important that enzymatically generated quinone derivatives react with chitosan in the heterogeneous system”: please rephrase. I agree about the difficulty of estimating the amount of amine groups accessible for reaction, but the authors could try at least with colorimetric tests, such as ninhydrin or salicylaldehyde.

Lines 258-259: “The reason for the use of a pH 6.0 buffer is simply because chitosan did not dissolve at pH 7.0”: and the reason for that is the pKa of chitosan, which is about 6.5.

Table 2: there is “chiton” instead of chitosan.

Figure 7a is not readable: please rescale it.

CONCLUSION: Line 315: the procedure is not “constructed”… please rephrase. Could the authors provide some figures of merit regarding the actual competitiveness (also in terms of economic factors) of their proposed method with other, commercially available, systems? In the simplest way, how well would their approach compare with activated charcoal in terms of efficiency and overall costs?

Author Response

Thank you for your reviewing. We corrected and prepared the R1 manuscript according to reviewer's comments.

Answers to the comments from reviewer #1

On correction of grammatical mistakes and spelling error

P1, L13: “enhanced with” changed to “increased with”.

P1, L15: “completely or highly removed” changed to “completely or mostly removed”.

P1. L35-38: The sentence was modified and written up.

P2, L69: The last sentence in the introduction was deleted.

P3, L98: “with a bullet for titration” changed to “with a bullet”.

P5, L202: The chitosan film was prepared according to the procedure described in our previous articles. The references are cited.

P1, L2-3: The title was changed according to reviewer’s comment.

On quinone adsorption

Quinone adsorption occurs by the chemical reaction between quinone derivatives enzymatically generated from alkylphenols and chitosan’s amino groups. The mechanism was already explained in detail by instrumental analysis (NMR and IR) in our and Payne’s previous articles. Therefore, the related references are cited (See p6, L190-192).

On the specific surface area

As we don’t have a BET surface analyzer, the specific surface areas of the chitosan powders used were calculated from the average diameter. Since the same explanation was made in our previous articles, the references were cited.

On adsorption on chitosan powders at pH 7.0

The pH value was adjusted to pH 7.0 to use chitosan powders in the heterogeneous systems, because chitosan is insoluble at pH 7.0. The explanation was modified (See p9, L246-249).

On homogeneous reaction at pH 6.0

The explanation on the homogeneous reaction at pH 6.0 was modified and the reference was cited (See p9, L254-256). In addition, chitosan is not soluble at pH 7.0, which is the optimum pH of PPO.

On Figure 7(a)

The scale of the y axis (0 to 0.40) was used in Figure 7. Since enzymatically generated was successfully adsorbed on chitosan beads and powders under these conditions, the absorbance at 400 nm was considerably repressed. We should show the repression of absorbance increase due to quinone adsorption. Therefore, this scale was used.

On the conclusion section

The conclusion sections were deleted and modified to shorten the conclusion section. We understand that it is important that some control experiments are performed with commercially available samples or activated charcoal. However, we didn’t perform these experiments. One reason is that the referee required to delete the description on the results of these results in previous reviews.

On the text

We a little shortened the text and added some explanation according to reviewer’s comment.

Reviewer 2 Report

Reviewer’s Comments to author. Alkylphenols are serious pollutants in aqueous solutions.  Removal of alkylphenols by adsorption techniques with combination of oxidoreductase, polyphenols oxidase and chitosan is a new topic. The present study is also interesting.  Nevertheless, the manuscript (MS) needs a major revision before considering for publication in Polymers. The authors have to address the following points: 1. Affiliation of author didnot follow the rule of Polymers. The authors have to correct it. 2. So many keyword in the MS. I think that 5 ones are enough. 3. Cited references are not good when many papers are cited in one sentence, for example [16-17, [20-23], [29-32],,… 4. The novelty of this work should be emphasize in Introduction. 5. Manuscript didnot follow the template of Polymers. The Figures and Tables must be put near the text. 6. Why pH 7.0 by phosphate buffer with ionic strength. The pH is important effect for adsorption of charged pollutants such as alkylphenols. The authors should refer to some papers: Polymers 10 (2), 220; Environmental Chemistry 14 (5), Materials 12 (3), 450; 7. Detail of UV-Vis and HPLC measurements should be added. 8. Adsorption of quinone onto chitosan beads are not clear. Since the quinone and chitosan are hydrophilic that the changes in surface charge before and after adsorption are important. Some papers are recommended for the authors to solve this matter: Colloid and Polymer Science 297 (1), 13-22; Colloid and Polymer Science 293 (7), 1877-1886; Journal of Chemistry 2017, Volume 2017, Article ID 1986071, 10 pages. 9. Line 421 didnot show the data that is not applicable. The author should add results 10. Two step procedure is used in this work. Is it similar to two-step adsorption model. The authors should read some papers about two-step adsorption model: Materials 12 (3), 450; International Journal of Polymer Science 2018, Article ID 2830286, 11 pages; Polymers 10 (2), 220; Environmental Chemistry 14 (5), 327-337; Colloid and Polymer Science 293 (7), 1877-1886. 11. Conclusions are too long so that I think it should be rewritten. 12. An important thing is that all Figures in this paper didnot show the error bar of replicated experiments. Did the author carry out experiments in triplicates.  If yes, error bars are required. If not, the experimental data are not convincing.

Author Response

Thank you for your reviewing. We corrected and prepared the R1 manuscript according to reviewer's comments.

Answers to the comments from reviewer #2

On the affiliation

The affiliations of the authors were revised according to the comment.

On the keywords

The keywords were decreased to 5 .

On the numbers of cited references in one sentence

Two or more references were cited in one sentence, because there are some or many related articles.

On the purpose of this study

The main purpose is described in the last sentences of the introduction section.

On Figures and Tables

Figures and Tables were inserted in the text. Tables were remade to insert them in the text.

On the use of a pH 7.0 buffer

The optimum pH value of PPO used in this study was determined to be 7.0 in our previous article. The explanation on the use of a pH 7.0 buffer was described (See p4, L131-134).

On adsorption or removal of alkylphenols

The principle of removal of alkylphenols in this study is the chemisorption of enzymatically generated quinone derivatives on chitosan beads or powders. This reaction is not based on the electrostatic interaction. In this study, the electrostatic interaction is not used for removal of alkylphenols. The mechanism of removal of ionic compounds referred by the reviewer is different from one in this study. Therefore, these articles were not cited. We hope that the reviewer understands our opinion.

On the measurement of UV-visible spectra and HPLC analysis

The measurement of UV-visible spectra is described in the 2.2 section and the related articles are cited. The HPLC analysis is described in the 2.3 section and the related articles are cited. Therefore, we didn’t describe additional explanations. We hope that the reviewer understand our opinion.

On mechanism of quinone adsorption on chitosan

Adsorption of enzymatically generated quinone derivatives onto chitosan beads was based on the chemical reaction of quinone derivatives with chitosan’s amino groups. Since this is a chemical reaction, this adsorption was referred to as the chemisorption. This term “chemisorption” has been used in the articles published by G. F. Payne and his group. Since the reaction mechanism is explained in detail, the related articles were cited. See p6, L190-192.

On the two-step reaction of adsorption in this study is based on the enzymatically generated quinone derivatives from alkylphenols and the subsequent nonenzymatic quinone adsorption on chitosan beads. This is described in this manuscript. In addition, this two-step reaction is different from one in the articles cited by the reviewers. Therefore, we didn’t cite the articles cited by the reviewers. We hope that the reviewer understands our opinion.

On rewriting of conclusion

The conclusion section was rewritten and shorten for readers to understand the conclusion.

On the error bar in Figures

We can understand the necessity of error bars in Figures. However, since many plots are shown in each graph for Figures 1 to 4, it becomes hard to see or understand the graphs. So, we didn’t insert the error bars. Due to the same reason, we didn’t insert the error bars in the figures in our previous articles.

Round  2

Reviewer 2 Report

The revised paper is suitable for publication.

Author Response

Thank you for your e-mail. We corrected and made the R2 manuscript according to reviewer’s comments. Reviewer’s comments helped us for accepting and publishing our study.

Answers to the comments from reviewer

On the molecular weight of the used chitosan

The nominal value of viscosity of the used chitosan is published. The viscosity of chitosan solution was described in place of molecular weight (See P2, L82-83).
